# *Corynebacterium accolens* Has Antimicrobial Activity against *Staphylococcus aureus* and Methicillin-Resistant *S. aureus* Pathogens Isolated from the Sinonasal Niche of Chronic Rhinosinusitis Patients

**DOI:** 10.3390/pathogens10020207

**Published:** 2021-02-14

**Authors:** Martha Alemayehu Menberu, Sha Liu, Clare Cooksley, Andrew James Hayes, Alkis James Psaltis, Peter-John Wormald, Sarah Vreugde

**Affiliations:** 1Department of Surgery-Otolaryngology, Head and Neck Surgery, The University of Adelaide, Basil Hetzel Institute for Translational Health Research, Central Adelaide Local Health Network, Woodville 5011, Australia; martha.menberu@adelaide.edu.au (M.A.M.); sha.liu@adelaide.edu.au (S.L.); clare.cooksley@adelaide.edu.au (C.C.); alkis.psaltis@adelaide.edu.au (A.J.P.); peterj.wormald@adelaide.edu.au (P.-J.W.); 2Department of Medical Microbiology, School of Biomedical and Laboratory Sciences, College of Medicine and Health Sciences, University of Gondar, Gondar 196, Ethiopia; 3Department of Microbiology and Immunology, Peter Doherty Institute for Infection and Immunity, University of Melbourne and The Royal Melbourne Hospital, Melbourne 3000, Australia; andrewhayes.bio@gmail.com

**Keywords:** chronic rhinosinusitis, *Corynebacterium accolens*, microbiota, sinus health

## Abstract

*Corynebacterium accolens* is the predominant species of the healthy human nasal microbiota, and its relative abundance is decreased in the context of chronic rhinosinusitis (CRS). This study aimed to evaluate the antimicrobial potential of *C. accolens* isolated from a healthy human nasal cavity against planktonic and biofilm growth of *Staphylococcus aureus* (*S. aureus*) and methicillin-resistant *S. aureus* (MRSA) clinical isolates (CIs) from CRS patients. Nasal swabs from twenty non-CRS control subjects were screened for the presence of *C. accolens* using microbiological and molecular techniques. *C. accolens* CIs and their culture supernatants were tested for their antimicrobial activity against eight *S. aureus* and eight MRSA 4CIs and *S. aureus* ATCC25923. The anti-biofilm potential of *C. accolens* cell-free culture supernatants (CFCSs) on *S. aureus* biofilms was also assessed. Of the 20 nasal swabs, 10 *C. accolens* CIs were identified and confirmed with *rpoB* gene sequencing. All isolates showed variable antimicrobial activity against eight out of 8 *S. aureus* and seven out of eight MRSA CIs. Culture supernatants from all *C. accolens* CIs exhibited a significant dose-dependent antibacterial activity (*p* < 0.05) against five out of five representative *S. aureus* and MRSA CIs. This inhibition was abolished after proteinase K treatment. *C. accolens* supernatants induced a significant reduction in metabolic activity and biofilm biomass of *S. aureus* and MRSA CIs compared to untreated growth control (*p* < 0.05). *C. accolens* exhibited antimicrobial activity against *S. aureus* and MRSA CIs in both planktonic and biofilm forms and holds promise for the development of innovative probiotic therapies to promote sinus health.

## 1. Introduction

Disruption of the human nasal microbiome homeostasis is found in patients with chronic rhinosinusitis (CRS). CRS is an inflammatory disorder of the mucosa of the nasal cavity and paranasal sinuses, characterized by various clinical manifestations including sinus/facial pain, nasal congestion, rhinorrhoea, post-nasal discharge, and a reduced sense of smell for a minimum of 12 weeks duration [1]. Whilst the aetiology of CRS is thought to be multifactorial, disruption of the microbial community residing in the sinuses, termed dysbiosis, has recently been implicated in CRS pathophysiology, in particular in more severe patients [2]. Dysbiosis is generally described as an imbalance of pathologic and commensal bacteria, which are involved in the protection against overgrowth of pathobionts or potentially disease-causing organisms [3,4]. In CRS patients, the microbiome is characterised by a decrease in the relative abundance of *Corynebacterium* and an expansion of pathogenic bacteria including *Staphylococcus*, *Haemophilus*, *Moraxella* and *Enterobacteriacea* [5]. Among pathogenic species, *Staphylococcus aureus* is the most frequently isolated species in patients with CRS [6]. Furthermore, exacerbations of CRS due to *S. aureus* and methicillin-resistant *S. aureus* (MRSA) have been reported in severe recalcitrant disease, leading to immune dysregulation, barrier dysfunction, biofilm formation, and worse clinical outcomes [7,8,9]. 

Several studies have shown that microbes that typically exist within the mucosa of the nasal cavity compete amongst themselves and inhibit the growth of competitors either by releasing antagonistic substances or by limiting access to nutrients from the surrounding environment [10,11]. For example, *Corynebacterium accolens*, a common commensal nasal species secretes the enzyme triacylglycerol lipase (LipS1) that degrades triacylglycerol to produce free fatty acids that interfere with the growth of *Streptococcus pneumoniae* in the nasal cavity [12,13]. 

Similarly, probiotic bacteria have been acknowledged as a potential novel treatment in various diseases of the gut linked to dysbiosis, as they can interfere with the growth of pathogenic organisms and provide a host-beneficial advantage [14]. In the context of CRS, manipulation of the sinonasal microbiome has been recognized as an innovative strategy to promote the re-establishment of sinonasal microbiome homeostasis and improve sinus health. As such, the potential of probiotic treatment in CRS has been demonstrated in in vivo models with several candidate bacterial species such as *S. epidermidis* and *Lactobacillus sakei* [15,16]. Similarly, intranasal administration of *Corynebacterium* species to carriers of *S. aureus* resulted in *the* eradication of *S. aureus* in >70% of carriers [17]. 

To design a probiotic therapy to combat dysbiosis and help shape the microbiome in the context of CRS, a well-tolerated, safe and effective cocktail of beneficial microbes with good antimicrobial activity against pathobionts is required. A recent international sinonasal microbiome study compared the sinonasal microbiome of 410 controls and CRS patients, in which *Corynebacterium* was the most prevalent genus present in >75% of CRS patients and controls with a significant reduction in its relative abundance in CRS patients compared to controls [5]. Together with the notion that *Corynebacterium* species can interfere with the growth of pathogens [12,13,17], these findings support the probiotic potential of *Corynebacterium* species. However, *Corynebacterium* species have also been reported to mediate sinusitis [16]. Therefore, it is important to define the commensal status of *Corynebacteria* at species and strain-level along with their interaction against the most prevalent CRS pathogens, particularly *S. aureus* and MRSA. 

This study was designed to isolate and characterize *Corynebacterium accolens* strains from healthy sinonasal cavities and evaluate their antimicrobial and antibiofilm potential against *S. aureus* and MRSA clinical isolates from CRS patients. 

## 2. Results

A total of 36 study subjects, 20 non-CRS controls (8 males and 12 females, aged between 20–70 years old) and 16 CRS patients (8 males and 8 females, aged between 36–90 years old) were included to collect nasal swab samples and identify the clinical isolates. The demographic characteristics and clinical data of each study subject are summarized in Table 1**.**

Based on the phenotypic API 20 Staph and chromogenic MRSA selective agar screening methods, 16 *S. aureus* isolates (8 methicillin-sensitive *S. aureus* (MSSA) and 8 MRSA) were identified from 16 CRS patients. Characteristics of *S. aureus* clinical isolates used in this study are shown in Appendix A.

### 2.1. Identification of C. accolens Isolates

From 20 non-CRS controls, 10 *C. accolens* isolates were identified by the API Coryne test kit with ≥90.0% similarity to known strains of *C. accolens* from the database. The molecular identification of all *C. accolens* isolates with the PCR amplification followed by gel electrophoresis resulted in a DNA fragment of approximately 446 bp in size (Appendix A). In order to confirm the strain level identification, partial *rpoB* gene sequencing was done for all *C. accolens* isolates. As shown in Table 2, the *rpoB* nucleotide sequence BLAST of 10 *C. accolens* isolates showed 96% to 100% similarity and 99% to 100% query coverage with the known culture collection strain, *C. accolens* CIP 104783 (from the Pasteur Institute Collection, Biological Resource Center of Pasteur Institute (CRBIP), Paris, France), GenBank accession number AY492242 identified previously [18]. 

### 2.2. Phylogenetic Relationship of the Strains

Comparison of the *rpoB* gene sequences with the corresponding *C. accolens* sequences from the GenBank database showed that *C. accolens* strains were placed in the evolutionary clade of *Corynebacterium* origin. Strains of *C. accolens*, C778 and C779 were clustered together with strains C784 and C785, respectively, with a bootstrap value of 100%. Furthermore, *C. accolens* strain C782 was clustered with a culture collection strain, *C. accolens* ATCC 49726 with a bootstrap value of 99%. The phylogenetic analysis based on the *rpoB* genes of all *C. accolens* clinical isolates and their closest related *Corynebacteria* species are indicated in Figure 1. 

### 2.3. Spectrum of Antimicrobial Activity

Eight methicillin-sensitive *S. aureus* (MSSA) and eight methicillin-resistant *S. aureus* (MRSA) were retrieved from the nasal cavity of CRS patients.

In the deferred growth inhibition assay all, of the *C. accolens* strains showed antagonistic effects against most MSSA and MRSA clinical isolates, but the degree of antagonism varied among the *C. accolens* strains. From all *S. aureus* clinical isolates, only one MRSA strain could not be inhibited by any of the *C. accolens* strains. Most of the *C. accolens* strains showed low inhibitory activities against various strains of MSSA, MRSA and reference strain *S. aureus* ATCC 25923 (inhibition zones of less than 5 mm). Interestingly, three of the isolated strains (*C. accolens* C779, *C. accolens* C781 and *C. accolens* C787), exhibited strong inhibition on the growth of MSSA C26 and MRSA C261 (inhibition zones of more than 8 mm) (Figure 2). *C. accolens* C781 was the most effective strain in inhibiting the growth of eight of eight (100%) MSSA and six of eight (75.0%) MRSA CIs tested. In contrast, *C. accolens* C782 was the least effective strain, showing inhibitory activities against only four of eight (50.0%) MSSA and two of eight (25.0%) MRSA CIs tested. Results are summarized in Table 3. 

### 2.4. Inhibitory Activity of C. accolens Concentrated Cell-Free Culture Supernatants (CFCSs) 

All of the *C. accolens* cell-free culture supernatants in the present study exhibited a significant dose-dependent antibacterial activity against all of the *S. aureus* isolates tested (MSSA C5, MSSA C26, MRSA C300, MRSA C261 and ATCC 25923) compared to the control group consisting of *S. aureus* in TSB without CFCS (*p* < 0.05). The highest concentration of 90% CFCS in TSB from all *C. accolens* strains significantly inhibited the planktonic growth of all *S. aureus* isolates (*p* < 0.0001) compared to untreated controls. However, lower concentrations of 30% CFCS in TSB (102 µg/mL) did not show antibacterial activity against any of the *S. aureus* isolates tested (*p* > 0.05) (Figure 3). 

*C. accolens* C779, C781 and C787 showed the highest antagonistic activity in the deferred growth inhibition assay, and their CFCSs demonstrated a strong anti-staphylococcal activity for the three *S. aureus* test strains (MSSA C26, MRSA C261 and ATCC 25923) and were selected for investigation of the dose-dependent growth inhibition of their CFCSs. The highest antimicrobial activity was observed for *C. accolens* C781 CFCS at all tested concentrations (30% to 90%) compared to controls (*p* < 0.05). *C. accolens* C779 and *C. accolens* C787 CFCSs showed significant growth inhibition at concentration of CFCS in TSB higher than 50% against MRSA C261 (*p* < 0.05) and 70% CFCS in TSB against MSSA C26 and *S. aureus* ATCC 25923 strains (Figure 4A–C). 

### 2.5. Characterization of the Inhibitory Effect of CFCSs Produced by C. accolens Strains

#### 2.5.1. Effect of Proteinase K and Heat Inactivation

The nature of the inhibitory substance produced by selected *C. accolens* strains was studied by treating their CFCSs with proteinase K and heat. CFCSs from the 3 selected *C. accolens* strains, C779, C781 and C787, at different concentrations completely lost their antimicrobial activity against the selected *S. aureus* strains after treatment with proteinase K (1 mg/mL) followed by heat (55 °C, 30 min) (Figure 4D–F). This indicated that inhibitory effects of the *C. accolens* strains were due to the proteinaceous nature of active substances. 

#### 2.5.2. Effect of Purified Protein Treatment 

Purified protein extracts from the selected *C. accolens* CFCSs showed concentration-dependent inhibitory activity against the tested *S. aureus* strains as indicated in Figure 4G–I (*p* < 0.05). Purified protein extracts from *C. accolens* strains C779 and C781 at 30% inhibited the growth of both MSSA C26 (*p* < 0.05), MRSA C261 and ATCC 25923 compared to control (*p* < 0.0001). Higher or 90% concentration of purified protein extracts from all tested *C. accolens* strains, C779, C781 and C787, exhibited a stronger antimicrobial effect (*p* < 0.0001) against all *S. aureus* strains compared to control. 

### 2.6. C. accolens CFCS Inhibits S. aureus and MRSA Biofilm Metabolic Activity 

The activity of *C. accolens* CFCS on the metabolic activity of 48-h biofilms formed by 3 representative *S. aureus* strains (MSSA C26, MRSA C261 and ATCC 25923) was evaluated using alamarBlue assays. As shown in Figure 5A–C, CFCSs obtained from *C. accolens* C779, C781 and C787 had a concentration-dependent reduction in metabolic activity of both MSSA C26 and MRSA C261 CIs in established biofilms with values reduced by 23% to 42% compared to respective positive control. Biofilm of *S. aureus* ATCC25923 could be inhibited by about 26% to 29% compared to control by *C. accolens* C781and C787 CFCS at concentrations ranging between 70% and 90%. However, only high concentrations of CFCS of 90% exhibited a significant inhibitory effect on biofilms formed by MSSA C26, MRSA C261 and *S. aureus* ATCC25923. The *C. accolens* CFCS exhibited different anti-biofilm activity against the 3 *S. aureus* strains tested. 

### 2.7. C. accolens CFCS Reduces S. aureus and MRSA Biofilm Biomass

Figure 5D–F show the effects of CFCSs extracted from *C. accolens* strains, C779, C781 and C787 on the *S. aureus* biofilm biomass established by clinical isolates MSSA C26 and MRSA C261 and reference strain ATCC 25923. Although the biofilm of MRSA C261 was less affected than MSSA C26, all tested *C. accolens* CFCSs at the highest concentration (90%) reduced the biofilm biomass of both *S. aureus* clinical isolates (between 28% and 40%). However, the *S. aureus* ATCC 25923 biofilm biomass was not affected by CFCS at any of the concentrations tested. 

## 3. Discussion

This study indicates the probiotic potential of *C. accolens* with the potential of this species to help shape a dysbiotic microbiome in the context of CRS by interfering with the growth of MSSA and MRSA in planktonic and biofilm form. Some beneficial nasal bacteria have been evidenced in providing beneficial functions to restore the sinonasal microbiome composition and improving immune health in patients with CRS through direct pathogen inhibition, secretion of a bioactive molecule or nutrient competition [19]. Our results show that *C. accolens* strains isolated from the sinonasal cavities of non-CRS control patients have antimicrobial activity against MSSA and MRSA strains isolated from the sinonasal cavities of CRS patients. Both MSSA and MRSA planktonic cells and biofilms were sensitive to *C. accolens* and our results indicate a secreted protein to likely be responsible for this activity. Although all *C. accolens* strains had anti-staphylococcal activity, there was a strain-dependent variability in the host range and strength of anti-microbial action. 

The human nasal cavity forms a complex microbial ecosystem colonized by several resident microorganisms comprising both commensals and pathobionts [20]. Emerging evidence indicates that *Corynebacteria* are the predominant genus in the sinonasal niche present in >75% of CRS patients and controls; however, the relative abundance of *Corynebacteria* is reduced in patients with CRS [5]. The reduction of *Corynebacteria*, and increased relative abundance of pathobionts such as *S. aureus* in these patients, reflect a potentially disturbed host–microbe–microbe balance that might contribute to the pathophysiology of this disease [2,5,21]. From those studies, it appears that *Corynebacteria* can be in general regarded as a commensal in the sinonasal cavities. This is also in line with our study where *C. accolens* was isolated from the sinonasal cavities of at least 50% of healthy controls. However, an outgrowth of *Corynebacterium* species has also been implicated in CRS [16]. Therefore, it is important to define the commensal status of *Corynebacteria* at the species level. Sequencing of *rpoB* and 16S rRNA genes are the most widely used molecular methods for reliable identification of *Corynebacterium* species, and the *rpoB* gene is considerably more polymorphic than the 16S rRNA gene for members of the genus *Corynebacterium* [18,22]. In this study, *rpoB* gene sequencing confirmed that all isolated *Corynebacterium* strains had a pairwise sequence similarity of 96% to 100% with a culture collection strain *C. accolens* CIP 104783, classifying them as *C. accolens*. Our phylogenetic analysis based on the *rpoB* gene sequences also revealed that some strains such as (C779 and C785), (C778 and C784) and (C781 and C783) were closely related and shared the same clade. Potentially due to their close phylogenetic relationship, these strains tended to have similar antimicrobial properties against *S. aureus* and MRSA.

In this study, all ten *C. accolens* strains were active against a variety of *S. aureus* and MRSA strains. Notably, three of the strains designed as C779, C781 and C787 showed strong inhibition against at least 6/8 (75%) MSSA and 5/8 (62.5%) MRSA CIs tested. In particular, *C. accolens* strain C781 had the widest host range and exhibited inhibitory activity against eight out of eight (100%) *S. aureus* and six out of eight (75%) MRSA CIs. Given that the antimicrobial properties appear similar in MSSA and MRSA, we speculate that the molecular mechanism is likely unrelated to known mechanisms of antibiotic resistance. An increasing amount of research has shown an inverse correlation between *Corynebacterium* and *S*. *aureus* nasal colonization [23,24,25]. For example, in a cohort of forty healthy adults, *C*. *accolens* negatively correlated with *S*. *aureus* colonization and positively correlated with *C*. *pseudodiphtheriticum* [23]. Moreover, a previous study by Uehara Y et al. described that frequently implanting *Corynebacterium* species eradicated *S. aureus* colonization in 12 of 17 healthy adult carriers, suggesting the beneficial role of *Corynebacterium* in the abolition of *S. aureus* nasal colonization [17]. Despite the complexity of *Corynebacterium*–*S. aureus* interactions and strain-level variations, those studies are in line with the present study and support the possibility of commensal *C. accolens* strains to be used as probiotic therapy in the context of CRS. 

Some studies have also focused on the activity of antibacterial products in commensal *Corynebacterium* CFCSs toward pathogens. For example, a secreted factor by *C. pseudodiphtheriticum*, a closely related *Corynebacterium* species to *C. accolens*, has revealed bactericidal activity against various *S. aureus* strains including MRSA [24]. In our study, the antimicrobial effects by *C. accolens* against *S. aureus* are at least in part due to a secreted antimicrobial substance. Moreover, given the abrogation of this effect by treatment of the CFCS with Proteinase K and heat, the bioactive product is likely a protein or peptide. The commensal-derived products in a complex sinonasal niche can directly act on challenging the pathogenic bacteria to maintain a well-balanced microbiome. Recently, a novel peptide antibiotic termed lugdunin produced by the nasal and skin commensal *Staphylococcus lugdunensis* has demonstrated strong bactericidal activity against *S. aureus* nasal and skin colonization as well as the immunomodulatory potential to protect the host [26]. 

The inhibitory activity of *C. accolens*, has been previously reported against *S. pneumoniae* and was mainly due to the production of primary triacylglycerol lipase and release of anti-pneumococcal free fatty acids from representative human nostril and skin surface triacylglycerols [13]. Furthermore, a previous study done by Ramsey MM et al. demonstrated another possibility of interaction between commensal *Corynebacterium* species and *S. aureus* pathobionts with a view to managing *S. aureus* nasal colonization. In this study, the virulence of *S. aureus* was heavily affected by commensal *C. amycolatum*, *C. accolens*, and *C. pseudodiphtheriticum* through altered expression of the *S. aureus* quorum sensing-controlled accessory gene regulator (*agr*) genetic locus involved in colonization and virulence, and shifting bacterial behavior from virulence to a commensal lifestyle [25]. It is particularly interesting to note that the inhibitory activity of commensal *C. accolens* strains in our study is more likely due to proteins affecting the growth of several MSSA and MRSA isolates that are pathogenic in CRS. However, more in-depth studies are needed to identify and characterize the *C. accolens* secreted protein that is responsible for the observed antimicrobial effect and to investigate the unexplored mechanism of action.

It is well known that nasal colonization with *S. aureus* along with MRSA, particularly in biofilm form, is associated with CRS disease recalcitrance and poor outcomes after sinus surgery [27,28,29,30]. Biofilms are thought to be the main mediators for disease persistence and treatment failure in various chronic disorders including CRS [30]. To our knowledge, no studies have investigated the anti-biofilm properties of commensal *Corynebacteria*, including *C. accolens*, against *S. aureus* and MRSA. However, Iwase and colleagues have previously shown the activity of another commensal nasal bacterium, *S. epidermidis*, in disrupting biofilm formation and previously established biofilms of *S. aureus* through the production of bioactive extracellular serine protease (Esp) [31]. In our study, CFCSs from selected *C. accolens* strains, C779, C781 and C787, showed a concentration-dependent inhibition of biofilms formed by *S. aureus* and MRSA CIs. Therefore, our findings support the potential use of *C. accolens* or bioactive compounds derived from those strains as antimicrobials against *S. aureus* biofilms. 

## 4. Materials and Methods

### 4.1. Collection of Clinical Isolates

Ethics clearance for the collection, storage and use of clinical isolates was obtained from TQEH Human Research Ethics Committee (HREC/15/TQEH/132). All study subjects provided their written consent to participate in this study. Nasal swabs were collected at the time of surgery from 16 CRS patients (*S. aureus* clinical isolates) and from 20 non-CRS control patients (*C. accolens* clinical isolates) in a sterile Amies transport medium (Sigma Transwab, MWE Medical Wire, Corsham, UK), placed on ice and immediately transported to our research laboratory for processing. 

*S. aureus* clinical isolates were identified from nasal swabs of CRS patients by culturing on mannitol salt agar (Oxoid, Basingstoke, UK) at 37 °C overnight followed by species-level identification using API 20 Staph test system (bioMerieux, Australia) according to the manufacturer’s instructions. All isolates were then screened for MRSA using a super sensitive and specific chromogenic MRSA selective agar (CHROMID^®^ MRSA SMART, bioMerieux, Australia) as described previously [32].

Non-CRS controls were patients undergoing septoplasty with no prior history of CRS, acute sinusitis, tonsillitis and ear infections in the 6 months prior to surgery. Nasal swabs in bacterial transport medium were first vortexed for 60 s and then diluted with phosphate buffer saline (PBS) 1:10. One-hundred-microliter aliquots of diluted samples were overlaid on Columbia agar plates with 5% sheep blood (Thermo Scientific, Oxoid, Australia) and incubated at 37 °C for 48–72 h. Cultures were inspected daily before colony identification, and visible bacterial colonies were subcultured onto tryptone soya agar (TSA) (Oxoid, Basingstoke, UK) supplemented with 0.8% Tween 80 and incubated for 48 h at 37 °C with 5% CO_2_ and screened phenotypically based on colony size and culture morphology. Biochemical characterization of the isolates was performed using the API Coryne test system (BioMérieux NSW, Australia) following the manufacturer’s instructions. The isolates were stored at −80 °C in tryptone soya broth (TSB) (Oxoid, Basingstoke, UK) plus 20% (*v*/*v*) sterile glycerol for further analysis.

### 4.2. C. accolens Genomic DNA (gDNA) Extraction and DNA Quality Control 

Bacterial gDNA was extracted from a 48hr culture suspension of the isolates using a DNeasy Blood and Tissue kit (Qiagen, Hilden, Germany), following the manufacturer’s recommendations. The concentration of DNA was determined by recording the absorbance at 260 nm (A_260_) using a NanoDrop 2000 spectrophotometer (Thermo Scientific, Waltham, MA, USA). The DNA purity was determined from the optical density absorbance value; A_260_/A_280_ nm ratio. Moreover, the DNA integrity was evaluated through gel electrophoresis. Briefly, 5 μL of each DNA extract was run on 1.8% agarose gel (Sigma-Aldrich, USA) in 1× Tris-Acetate-EDTA (TAE) buffer at 100 Volts for approximately 60 min and stained with 10,000× concentrate SYBR Safe (Invitrogen, Thermo Scientific, city, Canberra, Australia). DNA bands were visualized using the ChemiDoc^TM^ Touch imaging system (Bio-Rad, NSW, Australia).

### 4.3. Polymerase Chain Reaction (PCR) Amplification of Partial rpoB Gene 

PCR was carried out in a T100^TM^ Thermal cycler (Bio-Rad, NSW, Australia) using oligonucleotide primers, C2700F and C3130R (Appendix A) according to a previously described protocol with little modification [18]. Briefly, amplification reactions were performed in a final volume of 50 μL containing 5 μL of 10× standard Taq Mg-free buffer, 6 μL of 25 mM MgCl_2_ solution, 1 μL of 10 mM dNTP mixture (dATP, dTTP, dGTP and dCTP), 0.25 μL of 5.000 U/mL *Taq* DNA polymerase (all from BioLabs inc., Rowley, MA, USA), 1 μL of 10 μM concentration of each forward and reverse primer (Integrated DNA Technologies, SA, Australia), 25.75 μL nuclease-free water and approximately 200 ng/μL of DNA adjusted to 10 μL with nuclease-free water per reaction. Thereafter, PCR mixtures were subjected to 35 cycles of denaturation at 94 °C for 30 s, primer annealing at 50.6 °C for 30 s, and extension at 72 °C for 2 min. A negative control (RNAse free water) and positive control (*C. accolens* ATCC49726, from American Type Culture Collection, Manassas, VA, USA) reaction were set up for every PCR experiment. 

Amplified PCR products were separated on a 1.8% agarose gel (Sigma-Aldrich, St. Louis, MO, USA) with 10 µL of 10,000× concentrate SYBR Safe DNA gel stain (Invitrogen, Thermo Scientific, Canberra, Australia) in 1xTris-Acetate-EDTA (TAE) buffer at 100 Volts for 60 min. The gels were visualized using ChemiDoc^TM^ Touch imaging system (Bio-Rad). The size of PCR products was estimated by comparison with a 1kb plus DNA ladder (BioLabs Ltd., Rowley, MA, USA). The primer sequences and amplicon size used for the detection of *Corynebacteria* are described in Appendix A. 

### 4.4. rpoB Gene Sequencing and Strain Identification of C. accolens

The amplified PCR products were purified from agarose gel using QIAquick Gel extraction kits (Qiagen GmbH, Hilden, Germany) following the manufacturer’s extraction protocol. The concentration, purity and integrity of the recovered DNA samples were assessed using the NanoDrop 2000 spectrophotometer (Thermo Scientific, Waltham, MA, USA) and through agarose gel electrophoresis as specified. The purified DNA was then sent to the Australian Genome Research Facility Ltd. (AGRF) for sequencing. All samples were prepared for sequencing following the guide to AGRF sequencing service for Purified DNA (PD) as follows: 10 pmol of a primer (Forward or Reverse) + 12–18 ng of purified DNA + sterile MilliQ water (in a total volume of 12 µL). All sequencing results were analysed by comparing with NCBI GenBank database using the Blast program (http://blast.ncbi.nlm.nih.gov/Blast.cgi) for strain identification. 

### 4.5. Phylogenetic Analysis

Based on the *rpoB* gene sequence data, a phylogenetic tree elucidating the relationships between the identified strains was constructed. The nucleotide sequences were aligned through ClustalW program using MEGA (version 7.0) software, and evolutionary analysis was conducted using the neighbour-joining method keeping 1000 bootstrap replications [33]. The analysis involved 19 nucleotide sequences.

### 4.6. Nucleotide Sequence Accession Numbers

The *rpoB* sequences of 10 *C. accolens* strains, C778, C779, C780, C781, C782, C783, C784, C785, C786 and C787 have been deposited in the GenBank database (https://www.ncbi.nlm.nih.gov/genbank, accessed on 30 December 2020), under the accession numbers MT856944, MT856945, MT856946, MT856947, MT856948, MT856949, MT856950, MT856951, MT856952 and MT856953, respectively. 

### 4.7. Deferred Growth Inhibition Assay 

The antagonistic activity of all *C. accolens* clinical strains and a culture collection strain, *C. accolens* ATCC 49726 was evaluated against *S. aureus* clinical isolates and *S. aureus* ATCC 25923 (from ATCC, Manassas, VA, USA) using deferred growth inhibition assays as described previously [34] with modifications. Briefly, a 48 h *C. accolens* culture (20 µL, approximately 10^8^ cells) in TSB (test inhibitor strains) was pipetted onto the centre of a TSA plate supplemented with 0.8% Tween 80 and incubated for 48 h at 37 °C with 5% CO_2_. Single colonies of a 24 h culture of *S. aureus* (competitor strains) were suspended in sterile 0.9% saline and standardized to McFarland units of 1.0 (approximately 3 × 10^8^ CFU/mL) followed by a dilution of 1:10 in TSB. Next, approximately 250 µL of diluted culture were sprayed over the entire agar surface previously spotted with *C. accolens* and then incubated for a further 18–24 h. After incubation, a photograph was taken and the extent of the growth inhibition zone around the *C. accolens* spot was calculated quantitatively by measuring the diameter of the inhibition zone in millimetres minus the diameter of the central spot of the inhibitor strain. The test was done in triplicate, and the average of the diameters of the inhibition zones was obtained. The extent of inhibition was scored based on the inhibition zone diameter result as – (0 mm), + (<5 mm), ++ (5–7 mm), +++ (8–10 mm) and ++++ (>10 mm).

### 4.8. Preparation of Concentrated Cell-Free Culture Supernatants (CFCSs) from C. accolens Strains 

*C. accolens* strains were individually grown in 10 mL TSB in a shaking incubator at 37 °C for 48 hrs. The CFCSs were obtained from 48 hr cultures of *C. accolens* in TSB by centrifugation (4000× *g*, 4 °C for 10 min) followed by filtration through 0.2 μm sterile syringe filter (Pall Life Sciences, UK). Next, supernatants were passed through 3-kDa filter concentrator (Pierce Protein Concentrator, Thermo Fisher Scientific, Rockford, IL, USA) using centrifugation at 4000× *g*, 4 °C for 1–2 h to collect secreted proteins as described previously [35]. The protein concentration was then determined using Quick Start Bradford Protein Assay Kit (Bio-Rad Laboratories, USA) according to the manufacturer’s protocol. All reactions were carried out in duplicate. Concentrated CFCS was then stored as single-use aliquots at −80 °C until use.

### 4.9. Assessment of Anti-Bacterial Activity Using Concentrated CFCS and Minimum Inhibitory Treatment 

The inhibitory activity of CFCS from *C. accolens* isolates was tested against representative *S. aureus* and MRSA isolates following a broth micro-dilution assay protocol as described earlier [36] with minor modifications. The concentrated CFCS were first diluted in various concentration ranges (30%, 50%, 70% and 90%) using TSB. Next, 198 μL of the diluted mixture was dispensed in 96-well microtiter plates (Life Sciences, Boca Raton, FL, USA) to make CFCS with final concentrations of 102, 170, 238 and 306 µg/mL. The inoculum was then prepared from all tested *S. aureus* isolates by suspending 18–24 h young colonies pre-cultured on TSA in 3 mL of sterile saline (NaCl 0.9% *w*/*v*) and adjusted to 0.5 McFarland turbidity standard (1.0–2.0 × 10^8^ CFU/mL). Following this, 2 μL of bacterial suspension was inoculated to each well, and plates were incubated at 37 °C for 24 h. Wells containing bacteria without supernatant that was grown with the corresponding volume of TSB and sterile TSB-containing wells were used as positive and negative growth controls, respectively. After incubation, bacterial growth was determined by measuring the optical density (OD) at 595 nm using a microplate absorbance reader (iMark™, Bio-Rad, Australia). The inhibitory activity of the supernatant was calculated by comparing OD values between treated and untreated wells. The minimum inhibitory treatment was determined for 3 selected *C. accolens* strains’ concentrated CFCSs (*C. accolens* C779, *C. accolens* C781 and *C. accolens* C787). The assays were performed in three replicates, and the antimicrobial activity results are expressed as mean (± standard error of the means). 

### 4.10. Proteinase K and Heat Inactivation of CFCSs

Inactivation experiments of CFCS were carried out using Proteinase K and heat treatment as previously described [37] with minor amendments. An aliquot of CFCS from selected *C. accolens* strains were treated with proteinase K (Sigma-Aldrich, St. Louis, MO, USA), at a final concentration of 1 mg/mL at 37 °C for 5 h. After incubation, the samples were subjected to heat treatment at 55 °C for 30 min to inactivate protease enzymes. Next, the samples were allowed to cool to room temperature for 15 min before application. The antimicrobial activity of samples was then tested against representative *S. aureus* and MRSA clinical isolates and the reference strain *S. aureus* ATCC 25923 using a micro-dilution method in 96-well microtiter plates as specified. Proteinase K was used alone in the corresponding dilution broth (TSB) as a positive control, and wells containing TSB alone were used as a negative control. Three experimental replicates were performed for each protein sample, and data are presented as mean ± SEM of the three experiments.

### 4.11. Protein Clean-Up from CFCS and Detection of Anti-Bacterial Activity 

To remove salts and ionic contaminants such as detergents, lipids and phenolic compounds from CFCS, we used a 2-D Clean-Up Kit (GE-Biosciences, Piscataway, NJ, USA) according to the manufacturer’s instructions. At completion of the washing steps, 50 µL of sterile MilliQ water was added to resuspend the protein pellet. Following this, the anti-bacterial activity of purified protein samples (cleaned CFCS in sterile MilliQ water and TSB at a ratio of 30%, 50%, 70% and 90% (*v*/*v*)) was tested against representative *S. aureus* and MRSA clinical isolates and *S. aureus* ATCC 25923 as specified. Controls were bacterial inoculum in TSB and sterile MilliQ water at identical volume ratios along with positive growth controls (bacterial inoculum in TSB) and a negative control (sterile TSB). Results are presented as the mean values ± SEM of three independent experiments for each sample.

### 4.12. Assessment of Anti-Biofilm Activity Using C. accolens CFCSs

#### 4.12.1. Determination of Biofilm Metabolic Activity 

To assess the ability of *C. accolens* CFCS to inhibit the metabolic activity of *S. aureus* biofilms, alamarBlue biofilm assay was carried out using clear-bottom black 96-well plates as described previously [38]. Briefly, overnight cultures of *S. aureus* isolates grown in TSA were transferred into a sterile glass tube of 0.9% saline and adjusted to 1.0 McFarland turbidity standard (approximately 3 × 10^8^ CFU/mL). Next, the suspension was diluted into TSB at 1:15 ratio, and 150 µL of the final suspension was transferred to flat-bottom black 96-well microtiter plates and incubated at 37 °C for 48 h in the dark on a rotating shaker to form biofilms. The wells were washed twice with 200 µL 1× PBS to remove planktonic cells and air-dried for 5–10 min. Subsequently, wells were filled with 180 µL of different concentrations of *C. accolens* CFCS diluted in TSB (30%, 50%, 70% and 90%) and incubated at 37 °C on a rotating shaker for 24 h in the dark. Wells were then washed twice with 200 µL 1× PBS and air-dried for 5–10 min. Next, plates were stained with 200 μL alamarBlue (Invitrogen, Thermo Fisher Scientific, Hillsboro, OR, USA) and incubated for 3–5 h at 37 °C on a rotating shaker. Wells containing bacterial culture without CFCS treatment and wells containing TSB without bacterial culture were included as a positive growth control and a sterility control, respectively. The fluorescence intensity of each well was then read every hour by a microplate reader FLUOstar OPTIMA (BMG LABTECH, Ortenberg, Germany) at a wavelength excitation 530 nm and emission 590 nm until maximum fluorescence was reached. Comparing the average fluorescence intensity (FI) of the growth control wells with that of the CFCS treated wells, the inhibition percentages (% inhibition) of metabolic activity was calculated by the following formula: [(FI growth control − FI CFCS treated)/FI growth control] × 100. This assay was performed in triplicate for each treatment. 

#### 4.12.2. Determination of Biofilm Biomass 

Forty-eight-hour *S. aureus* biofilms treated with CFCS were washed twice with 1xPBS to remove the planktonic cells. The plates were then air-dried for 10 min, and the surface-attached biofilms were stained with 180 μL of 0.1% (v/v) crystal violet per well and incubated at room temperature for 15 min. Subsequently, the crystal violet was removed, and the plates were washed three times with 200 μL per well sterile MilliQ water to remove the unabsorbed stain. Next, 180 μL per well 30% acetic acid was added and incubated on a plate shaker until the crystal violet solubilised. Stained biofilm biomass was determined by measuring absorbance at 595 nm using the microplate reader (iMark™, Bio-Rad, NSW, Australia). All experiments included a sterility control well containing TSB without bacterial culture and a growth control well (as 100% cell mass) containing bacterial culture without CFCS treatment. The mean absorbance (Abs 595 nm) of the samples was determined, and the percentage of biofilm biomass reduction by the CFCS was calculated by the following formula: [(Abs growth control − Abs CFCS treated)/Abs growth control] × 100. All experiments were performed in triplicate and the mean value was calculated with the standard error. 

### 4.13. Statistical Analysis

All the measurements were performed in triplicate, and the values were expressed as mean ± standard error of the mean (SEM). The mean differences in absorbance value between CFCS treated and growth control wells were compared and analysed by One-way analysis of variance (ANOVA) using Dunnett’s multiple comparisons test for anti-bacterial and anti-biofilm assays. All experimental data analyses were performed in GraphPad Prism software version 8.0 (GraphPad Software, San Diego, CA, USA). Statistical significance was determined at *p*-value < 0.05. 

## 5. Conclusions

Taken together, the antimicrobial activity of *C. accolens* strains and their secreted proteins against *S. aureus* and MRSA clinical isolates in planktonic and biofilm form could be useful in the prevention of *S. aureus* outgrowth in the nasal microbiota and opens the possibility for a protective use of *Corynebacteria* against antibiotic-resistant *S. aureus* nasal colonization in a complex niche. Our findings have potential clinical implications towards the development of personalized probiotic therapy and might contribute to shaping the disrupted nasal microbiota in CRS.

## Figures and Tables

**Figure 1 pathogens-10-00207-f001:**
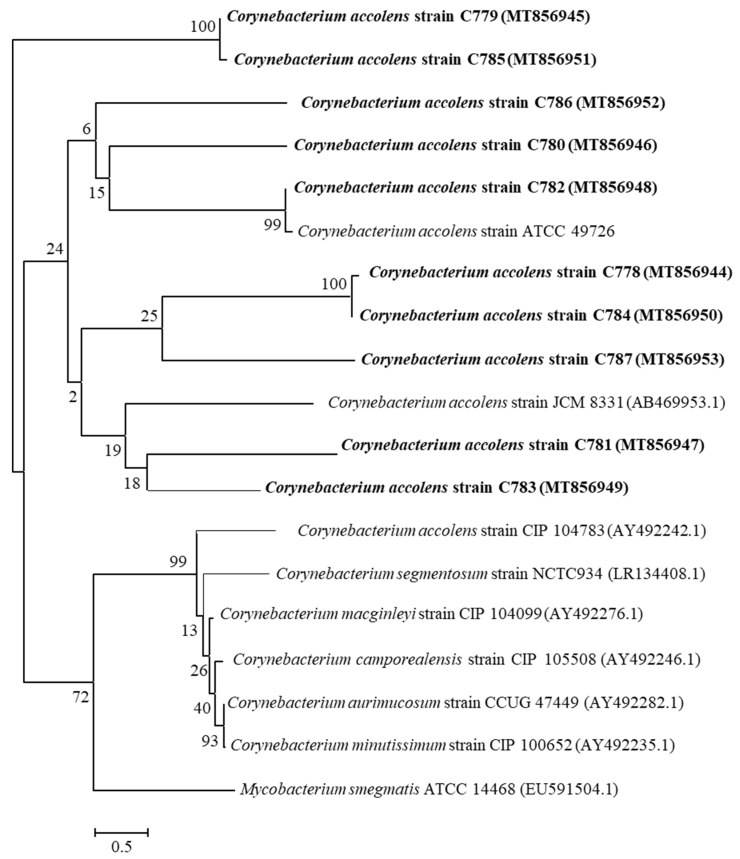
Phylogenetic tree showing the evolutionary relationships between 10 *C. accolens* nasal isolates, a reference strain *C. accolens* ATCC49726 and the type strains of related species (*C. accolens* CIP 104783, *C. accolens* JCM 8331 and other *Corynebacteria* species) based on *rpoB* gene sequences analysed using the neighbour-joining method. The percentages of replicate trees in which the associated taxa clustered together in the bootstrap test (1000 replicates) are shown next to the branches. Numbers in parentheses represent the sequence accession number in GenBank. *Mycobacterium smegmatis* ATCC 14468 was used as an outgroup. The scale bar represents 0.5-nucleotide substitutes per position.

**Figure 2 pathogens-10-00207-f002:**
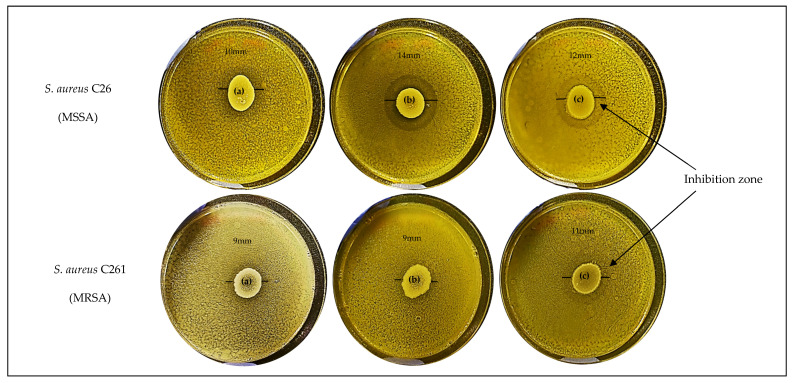
Antagonistic activity of selected *C. accolens* nasal isolates (**a**) *C. accolens* C779, (**b**) *C. accolens* C781 and (**c**) *C. accolens* C787 spotted on a lawn of *S. aureus* clinical isolates, MSSA C26 (top image) and MRSA C261 (bottom image) on tryptone soya agar (TSA) medium. The inhibition zone diameter was measured in at least three replicate experiments and the mean values were taken to score the extent of inhibition. The single line represents the growth inhibition zone. CI, Clinical isolate; MSSA, methicillin-sensitive *S. aureus*; MRSA, methicillin-resistant *S. aureus*.

**Figure 3 pathogens-10-00207-f003:**
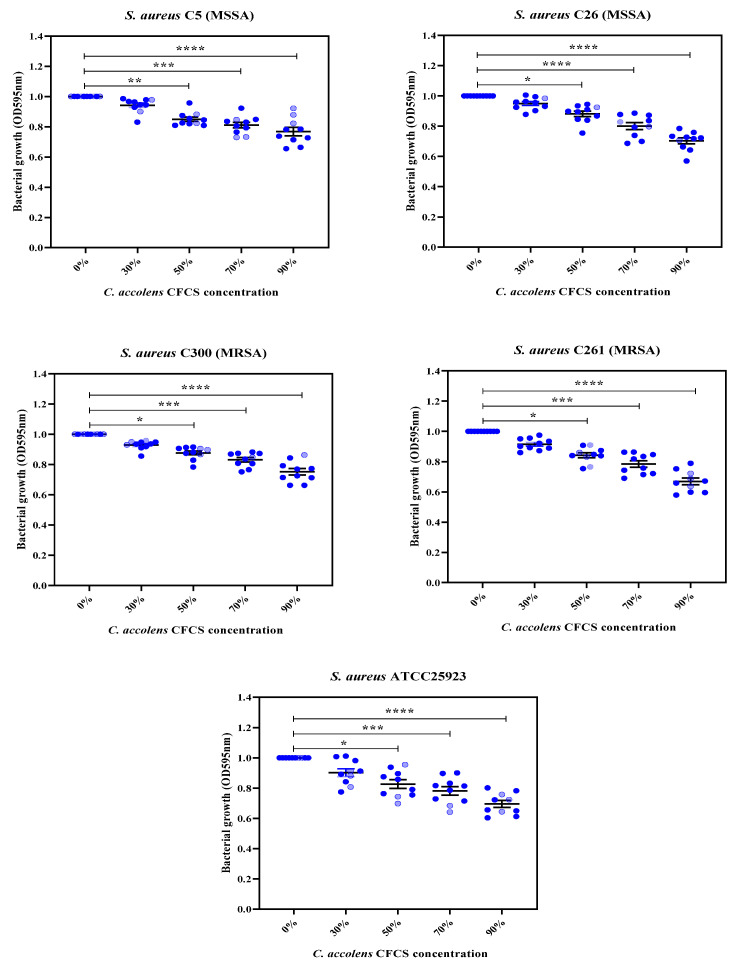
Growth inhibitory effect of *C. accolens* CFCSs (n = 10, represented by blue dots) against *S. aureus* (2 MSSA (C5, C26), 2 MRSA (C300, C261) and *S. aureus* ATCC25923) at various concentrations, 30%, 50%, 70% and 90% (CFCSs diluted in TSB), compared to controls (corresponding volume of TSB + SA, normalized to 1.0 OD value) following 24 h treatment. The results are expressed as means ± SEM of three independent experiments. SA, *S. aureus*; CI, clinical isolate; One-way ANOVA * *p* < 0.05; ** *p* < 0.01; ****p* < 0.001; **** *p* < 0.0001; SEM, standard error of the means.

**Figure 4 pathogens-10-00207-f004:**
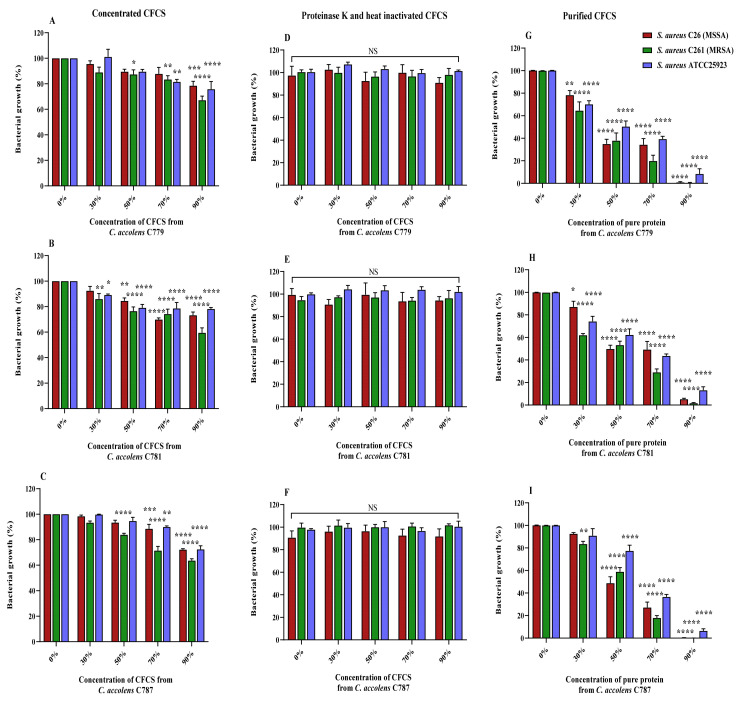
Antibacterial potential of *C. accolens* purified CFCSs (extracted from strains C779, C781 and C787) against *S. aureus* planktonic cells [MSSA C26 (red), MRSA C261 (green) and ATCC25923 (blue)]. (**A**–**C**) represents treatment of *S. aureus* strains with concentrated CFCSs from *C. accolens* strains C779 (**A**), C781 (**B**) and C787 (**C**) at concentrations of 30–90% or a positive control (corresponding bacterial inoculum in TSB); (**D**–**F**) represents treatment of *S. aureus* strains with proteinase K and heat inactivated CFCSs from *C. accolens* strains C779 (**D**), C781 (**E**) and C787 (**F**) at concentrations of 30–90% or a positive control (corresponding bacterial inoculum in concentrated TSB treated with proteinase K and heat); (**G**–**I**) represents treatment of *S. aureus* strains with purified CFCSs from *C. accolens* strains C779 (**G**), C781 (**H**) and C787 (**I**) at concentrations of 30–90% or a positive control (corresponding bacterial inoculum in TSB and MilliQ water, normalized to the water-diluted TSB). Bacterial growth (%) was determined as follows: [(Abs growth control—Abs CFCS treated)/Abs growth control] × 100; where, Abs = mean absorbance. Data presented as means ± SEM of three independent experiments. PK, proteinase K; CTSB, concentrated tryptone soya broth; SA, *S. aureus*; CI, clinical isolate; One-way ANOVA * *p* < 0.05; ** *p* < 0.01; *** *p* < 0.001; **** *p* < 0.0001; NS, not significant; SEM, standard error of the mean.

**Figure 5 pathogens-10-00207-f005:**
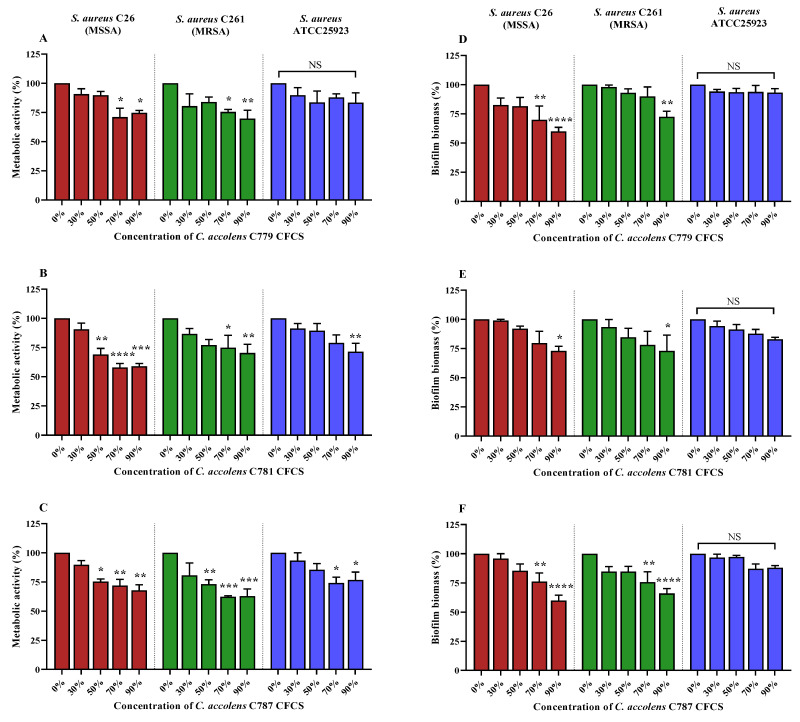
Anti-biofilm potential of *C. accolens* CFCS (extracted from strains C779, C781 and C787) on *S. aureus* biofilms established by clinical isolates (MSSA C26 (red), MRSA C261 (green) and ATCC25923 (blue)). (**A**–**C**) represent reduction of metabolic activity of biofilms formed by *S. aureus* strains (C26, C261 and ATCC25923) normalised to positive control (TSB + corresponding bacterial inoculum) in the presence and absence of CFCS from *C. accolens* strains, C779 (**A**), C781 (**B**) and C787 (**C**) diluted with TSB at different concentrations (30%, 50%, 70% and 90%). (**D**–**F**) represent biofilm biomass reduction of *S. aureus* strains (C26, C261 and ATCC25923) normalised to positive control (TSB + corresponding bacterial inoculum) after 24 hrs incubation with CFCS at different concentrations (30%, 50%, 70% and 90%) extracted from 3 *C. accolens* strains, C779 (**D**), C781 (**E**) and C787 (**F**). Values represent the means  ±  SEM of at least three independent experiments. Metabolic activity (% inhibition) = ((FI growth control − FI CFCS treated)/FI growth control) × 100, where FI = average fluorescence intensity, and Biofilm biomass (% reduction) = ((Abs growth control—Abs CFCS treated)/Abs growth control) × 100; where, Abs = mean absorbance. SA, *S. aureus*; CI, clinical isolate; One-way ANOVA * *p* < 0.05; ***p* < 0.01; *** *p* < 0.001; **** *p* < 0.0001; NS, not significant; SEM, standard error of the mean.

**Table 1 pathogens-10-00207-t001:** Demographic and clinical characteristics of the study subjects.

Characteristic	Non-CRS Controls, No. (%)	Patients with CRS, No. (%)
Number of subjects	20	16
Mean age (years)	45.7	64.0
Gender (M/F)	8/12	8/8
Active smoker	0 (0)	1 (6.3)
Asthma	6 (30)	8 (50)
Diabetes mellitus	1 (5)	0 (0)
Cystic fibrosis	0 (0)	0 (0)
GERD	6 (30)	3 (18.8)
Aspirin sensitivity	0 (0)	3 (18.8)
Tonsillitis in the past 6 months	0 (0)	0 (0)
Ear infection in the past 6months	0 (0)	0 (0)
Nasal polyposis	0 (0)	7 (43.8)

Abbreviations: CRS, chronic rhinosinusitis; F, female; GERD, gastroesophageal reflux; M, male.

**Table 2 pathogens-10-00207-t002:** Identification of *Corynebacterium accolens* using API Coryne 20 test system and *rpoB* gene sequencing.

Isolate Code	API Coryne 20 Identification ^†^ (% Similarity)	*rpoB* Gene Sequence Identification			
		Strains	% Similarity	% Query Coverage	Accession Number
C778	*C. accolens* (90.0)	*C. accolens*	98.3	100	MT856944
C779	*C. accolens* (95.6)	*C. accolens*	96.0	100	MT856945
C780	*C. accolens* (90.0)	*C. accolens*	97.6	100	MT856946
C781	*C. accolens* (99.4)	*C. accolens*	98.7	100	MT856947
C782	*C. accolens* (95.6)	*C. accolens*	99.5	100	MT856948
C783	*C. accolens* (90.0)	*C. accolens*	98.2	99	MT856949
C784	*C.* accolens (91.4)	*C. accolens*	98.3	100	MT856950
C785	*C. accolens* (90.0)	*C. accolens*	96.6	100	MT856951
C786	*C. accolens* (90.0)	*C. accolens*	97.3	100	MT856952
C787	*C. accolens* (90.0)	*C. accolens*	96.4	100	MT856953

*Note:***^†^** Results were interpreted based on various biochemical reactions on the API Coryne test strip and % similarity of the isolates were identified by comparing with *C. accolens* isolates deposited from the database (V4.0) using the apiweb^TM^ software.

**Table 3 pathogens-10-00207-t003:** Antagonistic activity of *C. accolens* against *S. aureus* clinical isolates in deferred growth inhibition assay.

Tested Pathogens	Diameter of Growth Inhibition Zone (mm) ^†^										
	Inhibitory Strains										
	*C. accolens* C778	*C. accolens* C779	*C. accolens* C780	*C. accolens* C781	*C. accolens* C782	*C. accolens* C783	*C. accolens* C784	*C. accolens* C785	*C. accolens* C786	*C. accolens* C787	*C. accolens* ATCC49726
**MSSA**											
*S. aureus* C329	−	+	++	+	−	++	++	−	+	+++	+
*S. aureus* C262	−	++	−	+	+	−	−	−	−	−	−
*S. aureus* C314	−	−	++	++	−	+	−	−	++	−	++
*S. aureus* C124	+	+++	+	+	+	−	++	+	−	++	+
*S. aureus* C5	+	+++	+	+	−	−	+	++	++	++	+
*S. aureus* C26	++	+++	++	++++	+	+++	++	+	+	++++	+
*S. aureus* C319	−	+	−	++	+	−	+	++	−	+	−
*S. aureus* C71	−	−	++	+	−	+	++	−	+	+	+
**MSSA (% inhibition)**	**3/8** **(37.5%)**	**6/8** **(75.0%)**	**6/8** **(75.0%)**	**8/8** **(100%)**	**4/8** **(50.0%)**	**4/8** **(50.0%)**	**6/8** **(75.0%)**	**4/8** **(50.0%)**	**5/8** **(62.5%)**	**6/8** **(75.0%)**	**6/8** **(75.0%)**
**MRSA**											
*S. aureus* C300	++	+++	++	+++	++	++	−	+	+	+++	++
*S. aureus* C310	+	−	+	+	−	++	++	+	+	++	+
*S. aureus* C292	++	+	+	++	−	−	+	−	+	+	+
*S. aureus* C295	−	−	−	−	−	−	−	−	−	−	−
*S. aureus* C261	+	+++	++	+++	+	++	+	++	++	++++	++
*S. aureus* C24	−	+	−	+	−	+	−	−	−	+	−
*S. aureus* C54	+	−	−	+	−	−	−	−	−	−	−
*S. aureus* C38	−	+	+	−	−	++	−	−	−	+	+
**MRSA (% inhibition)**	**5/8** **(62.5%)**	**5/8** **(62.5%)**	**5/8** **(62.5%)**	**6/8** **(75.0%)**	**2/8** **(25.0%)**	**5/8** **(62.5%)**	**3/8** **(37.5%)**	**3/8** **(37.5%)**	**4/8** **(50.0%)**	**6/8** **(75.0%)**	**5/8** **(75.0%)**
*S. aureus* ATCC25923	+	+	−	+	+	−	+	+	+	+	+
**Total** **(% inhibition)**	**9/17** **(52.9%)**	**12/17 (70.6%)**	**11/17 (64.7%)**	**15/17 (88.2%)**	**7/17** **(41.2%)**	**9/17** **(52.9%)**	**10/17** **(58.8%)**	**8/17** **(47.1%)**	**10/17 (58.8%)**	**13/17 (76.5%)**	**12/17 (70.5%)**

Note: **^†^** The extent of inhibition was scored based on the inhibition zone diameter result as: − (0 mm), + (<5 mm), ++ (5–7 mm), +++ (8–10 mm) and ++++ (>10 mm).

## Data Availability

The data presented in this study are available on request from the corresponding author.

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
