# Peer review of "Corynebacterium accolens Has Antimicrobial Activity against Staphylococcus aureus and Methicillin-Resistant S. aureus Pathogens Isolated from the Sinonasal Niche of Chronic Rhinosinusitis Patients"

_pathogens, 2021, doi:10.3390/pathogens10020207_

Round 1

Reviewer 1 Report

In their study "Corynebacterium accolens has Antimicrobial Activity against 2 Staphylococcus aureus and Methicillin-Resistant S. aureus Path-3 ogens Isolated from the Sinonasal Niche of Chronic Rhinosi-4 nusitis Patients" Martha Alemayehu Menberu et al. analysed the anti-microbial activity of C. accolens CIs and their culture supernatants against 8 S. aureus and 8 MRSA CIs and S. aureus ATCC25923.

Overall, their study is very well written and understandable. Also, the scientific importance is quite intersting and comprehensible.

Author Response

Thank you for your time in reviewing our manuscript.

Reviewer 2 Report

Manuscript “Corynebacterium accolens (CA) has Antimicrobial Activity against Staphylococcus aureus and Methicillin-Resistant S. aureus Pathogens Isolated from the Sinonasal Niche of Chronic Rhinosinusitis Patients” reports the antimicrobial activities of C. accolens against S. aureus strains including some MRSA. Though authors presented a very good set of data which clearly indicated that numerous CA could secrete proteinous substance which is inhibitory to MSSA and MRSA strains along with it’s antibiofilm activities, this manuscript is a hybrid of epidemiology and screening data. This makes bit confusing to the reader as they get lost whether the main aims of this study are: isolation and identification of Corynebacterium accolens or S. aureus strains (MSSA/ MRSA) strains or the antimicrobial synergy of these population dynamics in Sinonasal Niche because of these secreted antimicrobial agents.

Major points:

  1. Lack of focus research: this study could be very valuable if it would have been focused research using:
    1. One isolate of CA with high antimicrobial activity (for e.g. C781 from Table 3) against one aureus and try to identify Corynebacterium accolens inhibitory protein and the gene involved.
    2. Or use known CA (ATCC49726) whose genomic sequence available and try to figure out the gene making transposon library.
    3. Since both ATCC strain showing similar pattern (ATCC49726 also secretes this inhibitory protein which inhibits ATCC25923) as isolates, these inhibitory activities are not specific to Sinonasal isolates. It looks like secretion of inhibitory proteins are intrinsic to all CA strains. PMID: 30578265 also suggested similar competition among Nasal populations.
    4. It would have been great data, if correlation has been established where occurrence of CA with high antimicrobial activities resulting the decrease in MRSA population in that particular niche.
  2. Based on the data Table 3, these antimicrobial proteins are coming from Chromosome not from plasmids as most of the CA are at least inhibitory to some strains of aureus, if it were from plasmid, bacteria without plasmid will be negative for this antimicrobial protein.
  3. 3 showed a very little impact in growth OD Even though it appeared as significant, OD decrease from 1.00 to around 0.8 with 90% CFCS concentration, which is decrease in 0.2 OD. This figure also needs a proper control, where PBS or solvent used to concentrate 90% (CFCS) will be replaced with equal volume of buffer. Sometime, when we increase volume of (CFCS) to make 90%, it dilutes the growth media hence decrease the final growth.
  4. Table 3, the antimicrobial activities also dependent on the aureus strains: MSSA C26 and MRSA C261 have high susceptibility to most of the tested CA strains (with multiple “+”). It would be interesting to know why these strains are more susceptible than ATCC25923.
  5. Figure 5C, red, 70% bar has two stars whereas 50% has two, looks like typo as 70% bar looks taller than 50% in that figure. If this calculation is based on control, it should be other way around.

Recommendations: Focus experiment and logical conclusion can improve this manuscript. This can be done by following two reformatting:

  1. Focus on microbiome dynamics and show that CA limits the population of aureus (MSSA and MRSA) in specific Niche which correlates higher the antimicrobial activities of CA strains lower the number of S. aureus.
  2. Focus on identification of inhibitory gene (if it is a novel gene or protein) and protein in CA strains or the target gene in aureus.

Author Response

Thank you for your comments and suggestions to our manuscript. We have provided responses to specific comments and queries in the attached file. 

Reviewer 3 Report

The manuscript submitted by Menberu et al. focuses on the study of clinical isolates of C. accolans, and their supernatants, as potential antimicrobial and probiotic treatment for patients with chronic rhinosinusitis (CRI). Particularly, they focused on the capability of these clinical isolates to counteract the outgrowth of S. aureus MSSA and MRSA strains isolated from CRI-patients both as planktonic cells and biofilms. The paper is overall well-written, and the experimental results are solid and adequately analyzed. These results clearly indicate the potentiality of their probiotic treatment for the above-mentioned purpose, which I found of particular interest given the current spreading of antimicrobial resistance among pathogens. However, I found few concerns, mostly regarding how the data were presented and discussed, that needs to be addressed before this paper could be accepted for publication. My general and specific comments are reported below:

GENERAL COMMENTS:

In the Abstract section, the authors partially failed to highlight both innovative aspects of this interesting work, such as their evaluation of C. accolens antimicrobial potential towards S. aureus biofilms, and the conclusions that they drew from this study. Thus, I suggest the authors to reformulate this section to better guide the reader throughout the manuscript.

In the Introduction section, the authors reported that the most common pathogenic bacteria present in CRS patients are Staphylococcus, Hae48 mophilus, Moraxella, and Enterobacteriacea; yet this work is focused only on S. aureus and MRSA S. aureus. Why did the authors limit their investigation on these bacterial strains? Is S. aureus the only antibiotic-resistant bacterium among pathogens commonly found in CRS patients?

Could the difference, in terms of demographic and clinical characteristics, between healthy and CRS patients be significative for the efficacy of the proposed treatment? For instance, the difference in the average age of subjects can influence the microbial communities of patients and, therefore, the treatment needed to counteract S. aureus presence? Why did the authors choose such heterogeneous groups? They should expand on these concepts.

Regarding the antagonistic activity of C. accolans isolates against S. aureus ones, I think the authors should better emphasize how 3 of the C. accolans isolates gave higher bacterial inhibition even compared to the ATCC reference strain, and they should also discuss the possible motivation behind this behavior.

Why did the authors choose to evaluate the inhibitory effect of C. accolans CFCS only against 5 S. aureus strains out of all the 16 isolates (plus the ATCC control one)? Shouldn’t the efficacy of these treatments be tested against S. aureus strains as different as possible?

In Figure 3, the authors reported the growth inhibition of S. aureus by means of C. accolans CFCS; however, any indication regarding which CFCS are reported is provided; if each data point refers to a different CFCS, the authors should highlight this aspect through color-coding or something like that. If this is not the case, they should clearly state which CFCS are considering and why.

Why did the authors limit the investigation of dose-dependent growth inhibition of CFCS only on MSSA C26 and MRSA 175 C261 S. aureus strains?

The authors should also better present some of the obtained data, such as the effect of purified protein against S. aureus (Subsection 2.5.2), which is, in the present form, poorly described. Same comment for the subsection 2.5.3 and 2.5.4.

Why did the CFCS have no effect on biofilm biomass on S. aureus ATCC 25923, which is in theory the positive control for these experiments? What is the difference between this strain, which is a MRSA, and those isolated in this work?

Why, at the beginning of Discussion (page 12, lines 243-247) the authors referred only to the antimicrobial potential of C. accolens towards S. aureus and MRSA strains? What about MSSA strains?

Overall, the Discussion section of this manuscript needs to be highly modified and improved. In the present form, this section is simply a report and a parallel with other studies reported so far, but any critical speculations or link outside these studies is provided by the authors. For instance, on page 13, lines 293-295, the authors suggest that the antimicrobial activity of CFCS is likely due to the secretion of a protein or a peptide, yet any references regarding the ability of Corynebacteria to produce such compounds is presented or related to this speculation. This comment applies even better for how to concern all the experiments reported from Subsection 2.4 onwards, for which any discussion is completely missing. However, being these results the actual core of this paper, the authors should properly address them and their possible interpretation in the Discussion section, linking the obtained data with, for instance, the different physiology of S. aureus strains (planktonic cells vs. biofilms), the diversity between the clinical isolates, etc… By not doing so, the Discussion remains highly superficial and lacking on different levels, which is too bad, giving the interesting findings of the authors.

In the Conclusions section, the authors claimed that the CFCS can prevent the nasal colonization of S. aureus strains; however, the data here reported do not support this statement, as no bactericidal activity by means of CFU/ml count was performed. Hence, I suggest the authors to modify their conclusions by indicating something like “to prevent S. aureus outgrowth in the nasal microbioma”.

The use (or lack) of punctuation should be revised by the authors. For example, when a list of objects (or anything else) is reported, a coma followed by “and” should be used before the last term in the list (e.g., including 40 sinus/facial pain, nasal congestion, rhinorrhoea, post nasal discharge, and a reduced sense 41 of smell for a minimum of 12 weeks duration). A coma is also generally used before “which”, while no coma is utilized for “that”. Moreover, I found the use of expressions such as “8/8, 6/8, etc…” very confusing, as they can be read as fractions instead of “6 out of 8 strains” as they should be. I suggest the authors to change this throughout the entire manuscript.

In several sections of the manuscript, the text formatting (i.e., font, size, and line spacing) is either wrong, following the journal template, or not uniform; please, revise it and change it accordingly.

SPECIFIC COMMENTS:

Abstract:

Lines 19-21: Please, change “from the healthy human nasal 21 cavity” with “from a healthy human nasal 21 cavity” and provide the extended name for S. aureus, being the first time that it appears in the text.

Lines 24-25 and 25-27: Please, reformulate these sentences to improve their clarity.

Introduction:

Page 1, line 40: Please, change “post nasal” with “post-nasal”.

Page 2, line 45: What are pathobionts?

Page 2, lines 52-55: Please, change this sentence as follows: “Several studies have shown that microbes typically existing within the mucosa of the nasal cavity compete amongst themselves and inhibit the growth of competitors either by releasing antagonistic substances or limiting access to nutrients from the surrounding environment [9, 10]”.

Page 2, line 60: Please, remove “approach” to avoid redundancies and change “…dysbiosis and can interfere…” with “…dysbiosis, as they can interfere…”

Page 2, line 64-65: Please, change “the potential for probiotic…” with “the potential of probiotic…” and modify the expression “several different candidate probiotic bacterial species” to improve the clarity of the sentence.

Page 2, lines 72-74: Please, reformulate this sentence to improve its clarity.

Results section:

Page 3, lines 93-94: Given that the table referring to S. aureus isolates is in the Supplementary file, I suggest the authors to briefly write something about these results in the main text (i.e., being identified both Methicillin-resistant and sensitive strains) to help the reader in understanding their results.

Subsection 2.1:

Page 3, lines 96-98: Please, reformulate this sentence to improve its clarity.

Subsection 2.3:

Page 5, lines 130-134: Please, reformulate this sentence (punctuation is missing) to improve its clarity.

Table 3, page 6: Please, check and change the text formatting and position to make it uniform within the entire table.

Figure 2: the figure has been so much “manipulated” (I think for light correction purposes) that the inhibition zones are really hard to identify; please change it.

Subsection 2.4:

Figure 3: I think using “TSB+SA” to indicate the absence of CFCS is somehow confusing; I suggest the authors to change it with “0%” to be more consistent with the other concentration points. Same comment for Figures 4 and 5.

Page 8-9, lines 178-179: Please, change “significant growth inhibition at higher concentration of 50% CFCS in TSB..” with “…significant growth inhibition at concentration of CFCS in TSB higher than 50%...”.

Figure 4: what does the label “Bacterial growth (%)” in the y-axis mean? I think the authors should expand on this in the caption, being Materials and Methods reported at the end of the manuscript.

Subsection 2.5.3:

Page 10, lines 213-216: Please, reformulate this sentence to improve its clarity.

Figure 5: what are the labels “metabolic activity (%)” and “biomass biofilm (%)” on y-axis representing? I suggest the authors to briefly explain these labels in the figure caption, since the Materials and Methods section is at the end of the manuscript.

Discussion:

Page 12, lines 238-239: Please, reformulate this sentence to improve its clarity.

Page 12, lines 251-254: Please, reformulate this sentence to improve its clarity and avoid redundancies.

Page 12, line 255: Please, change the expression “…in these patients reflect…” by adding the required coma “…in these patients, reflect…”

Page 12, line 257: Please, change the expression “…it appears that in general Corynebacteria can be regarded…” with “…it appears that Corynebacteria can be in general (or generally) regarded…”.

Page 12, lines 258-259: Please, substitute “could be” with “was”; you did isolate those strains, so there is no need to use a conditional form of the modal “can”.

Page 12, lines 259-261: Please, reformulate this sentence, as it is identical to that in the Introduction section.

Materials and Methods:

Page 14, line 340: What do the authors mean with “heavy suspension of the isolates”? Can you be more specific regarding growth phase, etc?

Page 16, line 459: If the results are represented as mean with SEM it should be specified by the authors, as in the previous paragraph, same for line 470.

Page 16, lines 475-478: Please, reformulate this sentence to improve its clarity.

Author Response

Thank you for your comments and suggestions to our manuscript. We have provided responses to general and specific comments and queries in the attached file. 

Round 2

Reviewer 2 Report

Authors still need to revision based on previous suggestions. This manuscript is good set of data but not properly arranged to derive the conclusion.

  1. What are the findings ? CA creates extracellular antimicrobials against SA, it does not look like novel finding.
  2. Does higher level producers decrease the load of SA in the particular niche ? Author did not derive this conclusion.
  3. Better to modify this manuscript a typing (epidemiological data)  paper where CAs are typed based on genotypic and phenotypic characterristics including the level of this antimicrobial activities against SA.

Author Response

Thank you for reviewing our manuscript. I will respond point-by-point to your comments and questions.

Authors still need to revision based on previous suggestions. This manuscript is good set of data but not properly arranged to derive the conclusion.

What are the findings ? CA creates extracellular antimicrobials against SA, it does not look like novel finding.

This is the first manuscript describing the potential for C. accolens secreted products to inhibit the growth of MSSA and MRSA clinical isolates from CRS patients in planktonic and biofilm form.

Does higher level producers decrease the load of SA in the particular niche ? Author did not derive this conclusion.

This is the subject of future in vivo studies and outside the scope of the current manuscript.

Better to modify this manuscript a typing (epidemiological data)  paper where CAs are typed based on genotypic and phenotypic characterristics including the level of this antimicrobial activities against SA.

The purpose of the genotypic analysis and typing was to confirm the corynebacterium at species level and to evaluate the relation between the various strains rather than it being a purpose of its own.

We hope the modifications to our paper in response to all 3 reviewers and editor are accepted and that this manuscript can be published in Pathogens. We would like to again thank the reviewer for spending time to review our paper.